# ABDUCTIVE LOGICAL REASONING
# ON KNOWLEDGE GRAPHS

## ABSTRACT

Abductive reasoning is logical reasoning that makes educated guesses to infer the most likely reasons to explain the observations. However, the abductive logical reasoning over knowledge graphs (KGs) is underexplored in KG literature. In this paper, we initially and formally raise the task of abductive logical reasoning over KGs, which involves inferring the most probable logic hypothesis from the KGs to explain an observed entity set. Traditional approaches use symbolic methods, like searching, to tackle the knowledge graph problem. However, the symbolic methods are unsuitable for this task, because the KGs are naturally incomplete, and the logical hypotheses can be complex with multiple variables and relations. To address these issues, we propose a generative approach to create logical expressions based on observations. First, we sample hypothesis-observation pairs from the KG and use supervised training to train a generative model that generates hypotheses from observations. Since supervised learning only minimizes structural differences between generated and reference hypotheses, higher structural similarity does not guarantee a better explanation for observations. To tackle this issue, we introduce the Reinforcement Learning from the Knowledge Graph (RLF-KG) method, which minimizes the differences between observations and conclusions drawn from the generated hypotheses according to the KG. Experimental results demonstrate that transformer-based generative models can generate logical explanations robustly and efficiently. Moreover, with the assistance of RLF-KG, the generated hypothesis can provide better explanations for the observations, and the method of supervised learning with RLF-KG achieves state-of-the-art results on abductive knowledge graph reasoning on three widely used KGs.

## 1 INTRODUCTION

Abductive reasoning is a form of reasoning that is concerned with the generation of explanatory hypotheses for observed phenomena (Haig, 2012). It is a powerful tool across various research domains. For instance, in cognitive neuroscience, reverse inference (Calzavarini & Cevolani, 2022), representing an instance of abductive reasoning, is a crucial inferential strategy used to infer the most likely cognitive processes involved based on the observed brain activation patterns. Similarly, in clinical diagnostics, it is also recognized as one of the most important forms of reasoning for studying cause-and-effect relationships (Martini, 2023). Beyond these applications, abductive reasoning assumes a significant role in the process of reasoning to hypotheses across humans, animals, and computational machines (Magnani, 2023).

A knowledge graph (KG) stores information about entities, like people, places, items, and their relations in graph structures. Meanwhile, KG reasoning is a type of reasoning that leverages these knowledge graphs to infer or derive new information (Zhang et al., 2021a; 2022; Ji et al., 2022). In recent years, various logical reasoning tasks are proposed over knowledge graph, for example, answering complex queries expressed in logical structure (Hamilton et al., 2018; Ren & Leskovec, 2020), or conducting mining over the KG to obtain logical rules (Galárraga et al., 2015; Ho et al., 2018; Meilicke et al., 2019).

However, abductive reasoning using structured knowledge from KG is important yet lacks exploration. Consider the example of observation $O_1$ in Figure 1. This observation depicts a user fol-

---

*Equal Contribution

| Observations (O) | Hypotheses (H) | Interpretations |
|---|---|---|
| $O_1$ = {Grant Heslov, Jason Segel, Robert Towne, Ronald Bass, Rashida Jones} | $H_1 = V_? : Occupation(V_?, Actor) \land Occupation(V_?, ScreeeWriter) \land BornIn(V_?, LosAngeles)$ | The actors and screenwriters born in Los Angeles |
| $O_2$ = {Ipad 1st Gen, Ipod touch 4th Gen, Apple TV 1st Gen} | $H_2 = V_? : Brand(V_?, Apple) \land ReleaseYear(V_?, 2010) \land \neg Type(V_?, Phone)$ | The Apple products released in 2010 that are not phones |
| $O_3$ = {Covid-19, Seasonal Flu, Dysmenorrhea} | $H_3 = V_?, \exists V_1 : HaveSymptom(V_?, V_1) \land RelievedBy(V_1, Panadol)$ | The disease whose symptoms can be relieved by Panadol |

Figure 1: This figure shows some examples of observations and inferred logical hypotheses, expressed with discrepancies interpretations.

lowing five celebrities, *Grant Heslov, Jason Segel, Robert Towne, Ronald Bass,* and *Rashida Jones* on a social media platform. Given this observation, the social network service provider is interested in using structured knowledge to explain the user's observed behavior. Suppose we have a knowledge graph like Freebase (Bollacker et al., 2008), which includes some basic information about these people. We expect a method to utilize the information to find a complex structured hypothesis to explain the observations. For example, the knowledge graph may suggest that they were all actors and screenwriters born in Los Angeles. These characteristics interpret the user's intentions and behaviors. This complex structured hypothesis can be expressed as a logical expression $V_? : Occupation(V_?, Actor) \land Occupation(V_?, ScreenWriter) \land BornIn(V_?, LosAngeles)$, where $Occupation$ and $BornIn$ are relations and $Actor$, $ScreenWriter$, and $LosAngeles$ are entities from the KG, and $\land$ represents the logical conjunction operator, meaning AND. Consider another example in Figure 1: a user may search for several items in an e-commerce platform and view a series of products as shown in $O_2$, and the service providers can use a knowledge graph to generate a structured hypothesis like $H_2$, namely, they are the $Apple$ products released in 2010 that are not $phones$. There is also a more complicated example in medical diagnostics. Suppose we want to describe the observation $O_3 = \{Covid19, SeasonalFlu, Dysmenorrhea\}$, we can explain it through the logical hypothesis $H_3 = V_?, \exists V_1 : HaveSymptom(V_?, V_1) \land RelevedBy(V_1, Panadol)$, which means they are diseases $V_?$ with symptom $V_1$, and $V_1$ can be relieved by $Panadol$. From a general perspective, the proposed problem is a process of abductive logical reasoning on knowledge graphs, as it aims to find hypotheses that can best explain given observation sets (Josephson & Josephson, 1996; Thagard & Shelley, 1997).

A straightforward solution to this reasoning task is to use a search-based method to look for possible hypotheses given an observation. However, there are two challenges in the approach. The first challenge is the incompleteness of KGs, and the searching-based methods on KGs are sensitive to the missing edges (Ren & Leskovec, 2020). The second challenge is the complexity of logically structured hypotheses. The search space for the search-based methods contains a combinatorial number of candidate hypotheses. Consequently, the search-based method cannot deal with the observations needing a complex hypothesis to explain.

To deal with these challenges, we propose using generative models to generate logical hypotheses for the given observations in a supervised learning setting. In doing so, we sample hypothesis-observation pairs from the observed knowledge graphs (Ren et al., 2020; Bai et al., 2023) and then use the teacher-forcing method to train a transformer-based generative model (Vaswani et al., 2017) to generate hypotheses conditioned on the given observations. However, supervised training only minimizes structural differences between the generated and reference hypotheses, and higher structural similarity does not guarantee a better explanation. In addition, the unsatisfactory performance of this method can also be observed from the experiment results in Section 4.4. To address this issue, we propose a method called reinforcement learning from the knowledge graph (RLF-KG) which leverages proximal policy optimization (PPO) (Schulman et al., 2017) to minimize the discrepancy between observation and the conclusion drawn from the generated hypothesis.

The effectiveness and efficiency of the proposed methods are evaluated on three widely used knowledge graphs, FB15k-237 (Toutanova & Chen, 2015), WN18RR (Toutanova & Chen, 2015), and DBpedia50 (Shi & Weninger, 2018). Experiment results show that our generation-based methods are able to consistently outperform the search-based method in two evaluation metrics on three datasets, indicating its superiority in conducting abductive reasoning on KGs. Our contributions can be summarized as follows:

- We propose the task of abductive logical reasoning on knowledge graphs, in which given an observation set of entities, the goal is to find the logical hypotheses that can best explain the observation.

- We propose to use a generation-based method to address the difficulties of the incompleteness of KG and the complexity of the logical hypotheses.

- We propose reinforcement learning from knowledge graph (RLF-KG) to further improve the hypothesis generation model by incorporating the feedback from KG to minimize the differences between the observations and the conclusions drawn from the generated hypotheses.

## 2 PROBLEM FORMULATION

In this task, a knowledge graph is denoted by a $\mathcal{G} = (\mathcal{V}, \mathcal{R})$, where $\mathcal{V}$ is the set of vertices and $\mathcal{R}$ is the set of relation types. A *relation type* $r :\in \mathcal{R}$ is a mapping from vertex pairs in the KG to Boolean values, describing whether there is an edge of the type $r$ connecting one vertex to another. Namely, $r : \mathcal{V} \times \mathcal{V} \to \{\texttt{true}, \texttt{false}\}$ is defined by $\forall u, v \in \mathcal{V} : r(u, v) = \texttt{true}$ if the directed edge $(u, r, v)$ from $u$ to $v$ of type $r$ exists in the KG and $\texttt{false}$ otherwise.

Abductive reasoning is a type of logical reasoning that involves making educated guesses to infer the most likely reasons to explain the observations (Josephson & Josephson, 1996; Thagard & Shelley, 1997). Here we use simple syllogisms to explain the connections and differences between deductive and abductive reasoning. In deductive reasoning, suppose we have a major premise $P_1$: *All men are mortal*, a minor premise $P_2$: *Socrates is a man*, then we can draw the conclusion $C$ that *Socrates is mortal*. This can be also expressed as $P_1 \wedge P_2 \to C$. Meanwhile, in abductive reasoning, we also start with a premise $P$: *All cats like catching mice*, and then we have some observation $O$: *Katty like catching mice*. The abduction gives a simple yet most probable hypothesis $H$: *Katty is a cat*, as an explanation. Different from deductive reasoning, the observation $O$ should be entailed by the premise $P$ and the hypotheses $H$, which can be expressed by the following expression: $P \wedge H \to O$.

We adopt the open-world assumption of KG (Drummond & Shearer, 2006), under which the missing edges from the KG are regarded as unknown instead of false. Meanwhile, a reasoning model can only access the observed knowledge graph $\mathcal{G}$. The reasoning model is evaluated based on a hidden knowledge graph $\bar{\mathcal{G}}$, which contains the observed knowledge graph $\mathcal{G}$.

Any set of entities $O \subset \mathcal{V}$ is called an *observation set*. For example, $O$ can be a set of name entities like $\{GrantHeslov, JasonSegel, RobertTowne, RonaldBass, RashidaJones\}$. Given this observation, an abductive reasoner is required to give the most likely logical hypothesis that can explain the observation $O$. For the above example, the expected hypothesis $H$ in natural language is that they are *actors* and *screen writers*, and they are also born in *Los Angeles*. Mathematically, the hypothesis $H$ can be expressed by a logical expression of the facts of the KG: $H(V) = Occupation(V, Actor) \wedge Occupation(V, ScreenWriter) \wedge BornIn(V, LosAngeles)$. Although in this example the logical expression only contains logical conjunction AND ($\wedge$), we consider the more general first-order logical form including exisential quantifiers, AND ($\wedge$), OR ($\vee$), and NOT ($\neg$). Formally, we define a logical *hypothesis* $H$ on a graph $\mathcal{G} = (\mathcal{V}, \mathcal{R})$ as a predicate of a variable vertex $V_?$, given in disjunctive normal form,

$$H_{\mathcal{G}}(V_?) = \exists V_1, V_2, \ldots, V_k : e_1 \vee e_2 \vee \cdots \vee e_n, \tag{1}$$

$$e_i = r_{i1} \wedge r_{i2} \wedge \cdots \wedge r_{im_i}, \tag{2}$$

where each $r_{ij}$ is of one of these forms: $r_{ij} = r(v, V)$, $r_{ij} = \neg r(v, V)$, $r_{ij} = r(V, V')$, $a_{ij} = \neg r(V, V')$, where the lowercase letter $v$ represents a fixed vertex, the uppercase letters $V, V'$ represent variable vertices in $\{V_?, V_1, V_2, \ldots, V_k\}$, $r$ is a relation type, and all these vertices and relations are in $\mathcal{G}$. Note that the same hypothesis can be defined on different KGs as long as the fixed nodes and relations involved also exist in those KGs. When the context is clear, or the hypothesis refers to a general one that is supposed to be examined in multiple KGs, we omit the subscript $_\mathcal{G}$ for simplicity.

The *conclusion* of the hypothesis $H$ on a graph $\mathcal{G}$, denoted by $[\![H]\!]_\mathcal{G}$, is defined as the set of all the entities that make the hypothesis true on graph $\mathcal{G}$, given by

$$[\![H]\!]_\mathcal{G} = \{V_? | H_\mathcal{G}(V_?) = \texttt{true}\}. \tag{3}$$

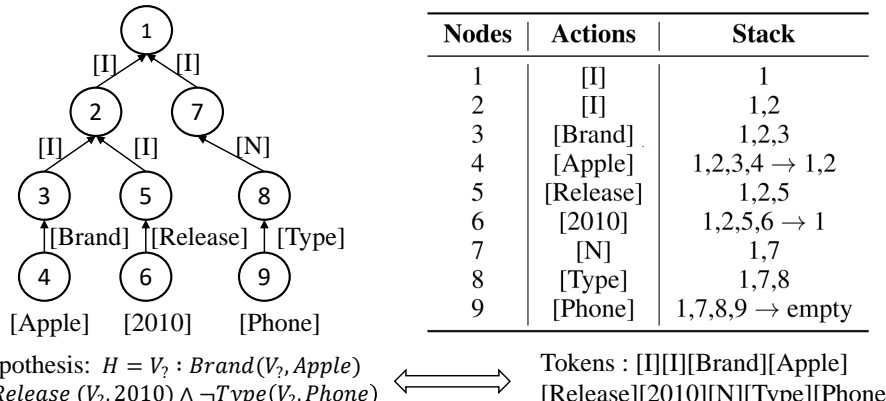

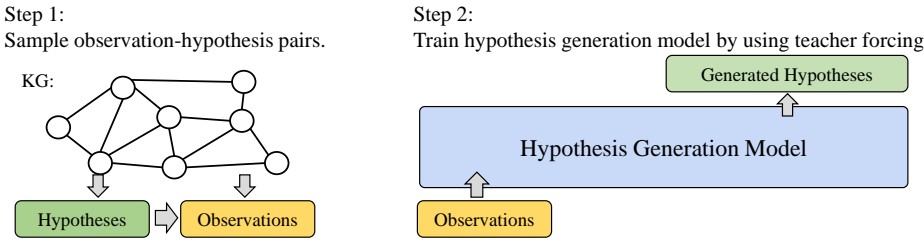

Figure 2: The illustration of tokenization of the hypothesis. We uniformly treated logical operations, relations, and entities as tokens, and created a correspondence between the hypotheses and a series of tokens. The details are described in Appendix A.

Figure 3: This figure shows the first two steps of training a hypothesis generation model. In Step 1, we sample some logical hypotheses with various patterns, use graph search on training graphs to acquire the training observations of these hypotheses, and then tokenize them. In Step 2, we train a conditional generation model to generate the hypotheses from observations.

Given an observation set $O = \{v_1, v_2, ..., v_k\}$, the goal of abductive reasoning is to find the hypothesis $H$ whose corresponding conclusion on the hidden graph $\bar{\mathcal{G}}$, $[\![H]\!]_{\bar{\mathcal{G}}} = O$, is as similar to $O$ as possible. Formally, we measure the similarity using the Jaccard index:

$$\texttt{Jaccard}([\![H]\!]_{\bar{\mathcal{G}}}, O) = \frac{|[\![H]\!]_{\bar{\mathcal{G}}} \cap O|}{|[\![H]\!]_{\bar{\mathcal{G}}} \cup O|}. \tag{4}$$

Then, the goal is to find a hypothesis $H$ that maximizes $\texttt{Jaccard}([\![H]\!]_{\bar{\mathcal{G}}}, O)$.

## 3 HYPOTHESIS GENERATION WITH RLF-KG

In this section, we present our method of generating a hypothesis for abductive logical knowledge graph reasoning. There are basically three steps. In the first step, we randomly sample the observation and hypothesis pairs from the knowledge graph and then tokenize them into sequences. In the second step, we use the sampled observation-hypothesis pairs as training data to train a generative model to generate the hypothesis from observations. The above two steps are illustrated in Figure 3. In the third step, we use RLF-KG to improve the generation model.

### 3.1 SUPERVISED TRAINING OF HYPOTHESIS GENERATION MODEL

In step 1, we randomly sample various types of hypotheses from the training knowledge graph and then conduct a graph search on the training graph to obtain the conclusion as the corresponding observation set of this hypothesis, as described in Appendix A

Then, we conduct tokenization of observations and hypotheses. Each of the entities in the observation is treated as a unique token, like `[Apple]` and `[Phone]` in Figure 2, and is associated

with an embedding vector. Without loss of generality, we sort the tokens for each observation in a pre-defined order so that various permutations of the same observation set are associated with the identical unique sequence of tokens.

The tokenization of the hypothesis is more complicated and is inspired by action-based parsing. We begin with utilizing the method proposed in other logical reasoning papers to convert the logical expression for a hypothesis into a directed acyclic graph (Hamilton et al., 2018; Ren & Leskovec, 2020; Ren et al., 2020), in which the logical operations such as intersection, union, and negation in the original logical structure are represented as edge attributes. Following previous work (Bai et al., 2023), we treat the logical operations of intersection, union, and negation as special tokens [I], [U], and [N] respectively. The relations and entities are treated as unique tokens like [Brand] and [Apple]. Then, we use a depth-first search algorithm described in Appendix A to acquire a sequence of *actions* representing both the content and the structure of the graph. This concludes the tokenization process for hypotheses. On the other hand, we can use the Algorithm 3 in Appendix A to recover a graph from an action sequence, which is regarded as the de-tokenization process of logical hypothesis.

In step 2, we train a generation model based on the sampled observation-hypothesis pairs. Suppose the token sequences for a pair of observation and hypothesis are $\mathcal{O} = [o_1, o_2, ..., o_m]$ and $\mathcal{H} = [h_1, h_2, ..., h_n]$, respectively. Then the loss for this example is defined to be the standard sequence modeling loss:

$$\mathcal{L} = \log \rho(\mathcal{H}|\mathcal{O}) = \log \Sigma_{i=1}^{n} \rho(h_i|o_1, o_2, ..., o_m, h_1, ..., h_{i-1}). \tag{5}$$

We use a standard transformer model to implement this model $\rho$ for conditional generations. There are two ways to use the conditional generation model. In the first approach, we use the encoder-decoder architecture from the original paper (Vaswani et al., 2017), in which we put the observation tokens as the input to the transformer encoder, and the shifted hypothesis tokens as the input to the transformer decoder for conditional generation. In the second approach, we concatenate the observation and hypothesis tokens and use the decoder-only transformer to generate the hypothesis tokens. We implement the two approaches, and both of them can be incorporated with the following RLF-KG method.

## 3.2 Reinforcement Learning from Knowledge Graph Feedback (RLF-KG)

After training the conditional generation model $\rho$, we try to improve it using reinforcement learning (Ziegler et al., 2020) with the feedback signals from the KG. Recall that, in the supervised training process, the model can only learn how to generate hypotheses structurally similar to the reference hypotheses. However, hypotheses with higher structural similarity do not necessarily guarantee logically better explanations. Motivated by this, we propose to use reinforcement learning from knowledge graph feedback (RLF-KG) to improve the trained conditional generation model $\rho$.

In step 3, we initialize the *model to be optimized* $\pi$ to be model $\rho$ resulting from supervised training and then fix $\rho$ as the *reference model*. Given the input observation $O$ and the generated hypothesis token sequence $\mathcal{H}$, we recover the corresponding hypothesis $H$ through the de-tokenization process and derive its conclusion on the training graph $\mathcal{G}$, namely $[\![H]\!]_{\mathcal{G}}$. Since $\mathcal{G}$ is the observed training graph, the model cannot acquire any information from the test edges. Therefore, the Jaccard similarity between $O$ and $[\![H]\!]_{\mathcal{G}}$ serves as an approximation, with no information leakage, to the objective of the abductive reasoning task defined in Equation 4. In view of the above, we choose this similarity to be the reward function $r$, hence introducing the feedback information from the training KG:

$$r(\mathcal{H}, \mathcal{O}) = \texttt{Jaccard}([\![H]\!]_{\mathcal{G}}, O) = \frac{|[\![H]\!]_{\mathcal{G}} \cap O|}{|[\![H]\!]_{\mathcal{G}} \cup O|}. \tag{6}$$

Again following (Ziegler et al., 2020) , we modify the reward by adding a KL divergence penalty to prevent the tuned model $\pi$ from producing excessively divergent hypotheses from the reference model. Then, we train the model $\pi$ using proximal policy optimization (PPO) (Schulman et al., 2017) with the expected modified reward on the training observation sets as the objective:

$$Objective(\pi) = \mathbb{E}_{\mathcal{O} \sim D, \mathcal{H} \sim \pi(\cdot|\mathcal{O})} \left[ r(\mathcal{H}, \mathcal{O}) - \beta \log \frac{\pi(\mathcal{H}|\mathcal{O})}{\rho(\mathcal{H}|\mathcal{O})} \right], \tag{7}$$

where $D$ the is training observation distribution and $\pi(\cdot|\mathcal{O})$ is the conditional distribution of $\mathcal{H}$ on $\mathcal{O}$ modeled by the model $\pi$. The process is summarized in Figure 4.

Step 3:
Optimize hypothesis generation model with Reinforcement Learning From Knowledge Graph feedback (RLF-KG).

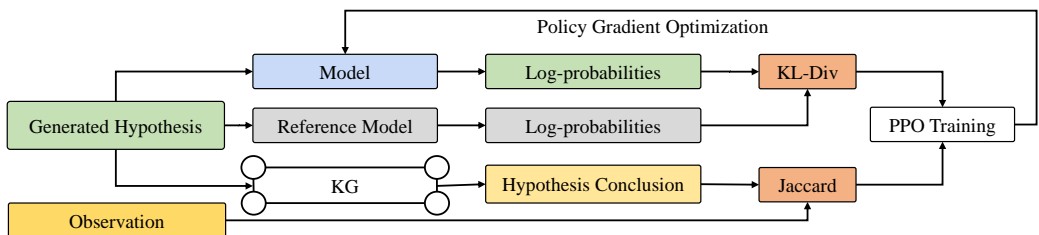

Figure 4: In Step 3, We use RLF-KG to encourage the model to generate hypotheses that draw conclusions more similar to given observations from KG.

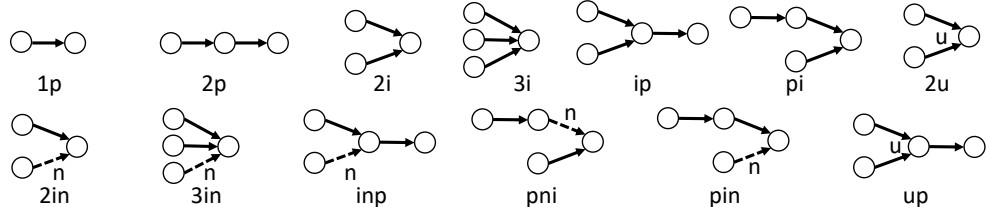

Figure 5: Thirteen types of logical hypotheses are considered in our task. Each of the hypothesis types is associated with one type of query graph that is used for sampling.

## 4 EXPERIMENT

### 4.1 DATASETS

We use three knowledge graphs, FB15k-237 (Toutanova & Chen, 2015), DBpedia50 (Shi & Weninger, 2018), and WN18RR (Toutanova & Chen, 2015). The number of training, evaluation, and testing edges and the number of nodes are reported in Table 1. We randomly separate the training, validation, and testing edges from these knowledge graphs with a ratio of 8:1:1. For each of these graphs, the training graph $\mathcal{G}_{train}$, validation graph $\mathcal{G}_{valid}$, and testing graph $\mathcal{G}_{test}$ are then the graphs induced by the training edges, training + validation edges, and training + validation + testing edges respectively.

Then, we sample pairs of observations and hypotheses as stated in Section 3.1. Meanwhile, we also impose some constraints on the samples. We restrict the observation sets to no more than thirty-two elements by sub-sampling the derived sets. Moreover, each validation hypothesis must have extra entities in the conclusion on the validation graph compared to the training graph, and each testing hypothesis must have extra entities in the conclusion on the testing graph compared to the validation graph. Following previous work on logical reasoning on KG (Ren & Leskovec, 2020; Ren et al., 2020), we chose to use thirteen pre-defined logical patterns to sample the hypothesis. Eight of them do not include negations, and are therefore called existential positive first order (EPFO) hypotheses: 1p/2p/2u/3i/ip/up/2i/pi. The other five involve negation, and we call them negation hypotheses: 2in/3in/inp/pni/pin. Note that the generated hypothesis may or may not in the same type as the reference hypothesis. The hypothesis structures are demonstrated in Figure 5. The numbers of samples drawn for each type can be found in Table 2.

### 4.2 EVALUATION METRIC

We use the objective of abductive reasoning stated in Section 2 as the main metric to measure the quality of the generated hypothesis. Suppose we have an observation $O$ and a generated hypothesis $H$. We first perform a graph search algorithm to find the conclusion of $H$ on the evaluation graph $\mathcal{G}_{test}$, $[\![H]\!]_{\mathcal{G}_{test}}$. Note that this graph contains ten percent of edges that are not seen during either the training or validation stages. Then, the Jaccard metric is given by

$$J_{test}(H, O) = \text{Jaccard}([\![H]\!]_{\mathcal{G}_{test}}, O) = \frac{|[\![H]\!]_{\mathcal{G}_{test}} \cap O|}{|[\![H]\!]_{\mathcal{G}_{test}} \cup O|}. \tag{8}$$

Table 1: The basic information about the three knowledge graphs used for the experiments, and their standard training, validation, and testing edges separation.

| Dataset | Relations | Entities | Training | Validation | Testing | All Edges |
|---|---|---|---|---|---|---|
| FB15k-237 | 237 | 14,505 | 496,126 | 62,016 | 62,016 | 620,158 |
| WN18RR | 11 | 40,559 | 148,132 | 18,516 | 18,516 | 185,164 |
| DBpedia50 | 351 | 24,624 | 55,074 | 6,884 | 6,884 | 68,842 |

Table 2: The detailed information for the queries used for training, validation, and testing.

| Dataset | Training Samples | | Validation Samples | | Testing Samples | |
|---|---|---|---|---|---|---|
| | Each Type | Total | Each Type | Total | Each Type | Total |
| FB15k-237 | 496,126 | 6,449,638 | 62,015 | 806,195 | 62,015 | 806,195 |
| WN18RR | 148,132 | 1,925,716 | 18,516 | 240,708 | 18,516 | 240,708 |
| DBpedia50 | 55,028 | 715,364 | 6,878 | 89,414 | 6,878 | 89,414 |

To investigate whether the generated hypothesis is similar to the reference hypothesis, we also propose to use SMATCH (Cai & Knight, 2013) to evaluate the similarity of hypothesis graphs. SMATCH is a metric originally proposed for evaluating the structural differences between two semantic graphs. As previous work points out, complex logical queries can be regarded as a special type of semantic graphs (Bai et al., 2023). We propose to use this well-established metrics to evaluate the structural differences between two hypotheses. The details of computing SMATCH on hypothesis graphs are explained in Appendix D. Here, we denote the SMATCH score as $S(H, H_{ref})$.

## 4.3 EXPERIMENT DETAILS

In this experiment, we use two transformer structures as the backbones of the generation model. For the encoder-decoder transformer structure, we use three encoder layers and three decoder layers. Each layer has eight attention heads with a hidden size of 512. Note that the positional encoding for the input observation sequence is disabled, as we believe that the order of the entities in the observation set does not matter. For the decoder-only structure, we use six layers, and the other hyperparameters are the same. In the supervised training process, we use AdamW optimizer and grid search to find hyper-parameters. For the encoder-decoder structure, the learning rate is 0.0001 with the resulting batch size of 768, 640, and 256 for FB15k-237, WN18RR, and DBpedia respectively. For the decoder-only structure, the learning rate is 0.00001 with batch-size of 256, 160, and 160 for FB15k-237, WN18RR, and DBpedia respectively, and linear warming up of 100 steps. In the reinforcement learning process, we use the dynamic adjustment of the penalty coefficient $\beta$ (Ouyang et al., 2022). More detailed hyperparameters are shown in Appendix F. All the experiments can be conducted on a single GPU with 24GB memory.

## 4.4 EXPERIMENT RESULTS

To prove the effectiveness of our proposed RLF-KG, we compare the Jaccard metric of the model before and after this process. The performance of all thirteen types of hypothesis types is shown in Table 3. In this table, we show the Jaccard index between the observations and the conclusions of the generated hypothesis drawn from the test graph. The models are evaluated on FB15k-237, WN18RR, and DBpedia50 respectively. On each dataset, we report the performance of the two transformer-based models under fully supervised training in the Encoder-Decoder row and the Decoder-only row. Meanwhile, we also report the performances when they cooperated with the reinforcement learning from knowledge graph feedback (RLF-KG).

We notice that the RLF-KG is able to consistently improve the performance of hypothesis generation on three different datasets. Meanwhile, RLF-KG can improve both encoder-decoder and decoder-only structured generation models. This improvement can be explained by the effectiveness of the RLF-KG method in incorporating the knowledge graph information into the generation model, rather than simplifying generating hypotheses that are similar to the reference hypothesis.

Additionally, after the RLF-KG training, the encoder-decoder model is better than the decoder-only structured transformer model. This can be explained by the nature of this task, as the task is required

Table 3: The detailed Jaccard performance of various methods.

| Dataset | Model | 1p | 2p | 2i | 3i | ip | pi | 2u | up | 2in | 3in | pni | pin | inp | Ave. |
|---|---|---|---|---|---|---|---|---|---|---|---|---|---|---|---|
| FB15k-237 | Brute-force Search | 0.980 | 0.563 | 0.639 | 0.563 | 0.732 | 0.633 | 0.744 | 0.585 | 0.659 | 0.479 | 0.607 | 0.464 | 0.603 | 0.635 |
| | Encoder-Decoder | 0.626 | 0.617 | 0.551 | 0.513 | 0.576 | 0.493 | 0.818 | 0.613 | 0.532 | 0.451 | 0.499 | 0.529 | 0.533 | 0.565 |
| | + RLF-KG | **0.855** | **0.711** | **0.661** | **0.595** | **0.715** | **0.608** | **0.776** | **0.698** | **0.670** | **0.530** | **0.617** | **0.590** | **0.637** | **0.666** |
| | Decoder-Only | 0.666 | 0.643 | 0.593 | 0.554 | 0.612 | 0.533 | 0.807 | 0.638 | 0.588 | 0.503 | 0.549 | 0.559 | 0.564 | 0.601 |
| | + RLF-KG | **0.789** | **0.681** | **0.656** | **0.605** | **0.683** | **0.600** | **0.817** | **0.672** | **0.672** | **0.560** | **0.627** | **0.596** | **0.626** | **0.660** |
| WN18RR | Brute-force Search | 0.997 | 0.622 | 0.784 | 0.776 | 0.955 | 0.666 | 0.753 | 0.605 | 0.783 | 0.757 | 0.762 | 0.540 | 0.630 | 0.741 |
| | Encoder-Decoder | 0.793 | 0.734 | 0.692 | 0.692 | 0.797 | 0.627 | 0.763 | 0.690 | 0.707 | 0.694 | 0.704 | 0.653 | 0.664 | 0.708 |
| | + RLF-KG | **0.850** | **0.778** | **0.765** | **0.763** | **0.854** | **0.685** | **0.767** | **0.719** | **0.743** | **0.732** | **0.738** | **0.682** | **0.710** | **0.753** |
| | Decoder-Only | 0.760 | 0.734 | 0.680 | 0.684 | 0.770 | 0.614 | 0.725 | 0.650 | 0.683 | 0.672 | 0.688 | 0.660 | 0.677 | 0.692 |
| | + RLF-KG | **0.821** | **0.760** | **0.694** | **0.693** | **0.827** | **0.656** | **0.770** | **0.680** | **0.717** | **0.704** | **0.720** | **0.676** | **0.721** | **0.726** |
| DBpedia50 | Brute-force Search | 0.997 | 0.705 | 0.517 | 0.517 | 0.982 | 0.461 | 0.783 | 0.754 | 0.722 | 0.658 | 0.782 | 0.544 | 0.700 | 0.702 |
| | Encoder-Decoder | 0.706 | 0.657 | 0.551 | 0.570 | 0.720 | 0.583 | 0.632 | 0.636 | 0.602 | 0.572 | 0.668 | 0.625 | 0.636 | 0.627 |
| | + RLF-KG | **0.842** | **0.768** | **0.636** | **0.639** | **0.860** | **0.667** | **0.714** | **0.758** | **0.699** | **0.661** | **0.775** | **0.716** | **0.769** | **0.731** |
| | Decoder-Only | 0.739 | 0.692 | 0.426 | 0.434 | 0.771 | 0.527 | **0.654** | 0.688 | 0.602 | 0.563 | 0.663 | **0.640** | 0.701 | 0.623 |
| | + RLF-KG | **0.777** | **0.701** | **0.470** | **0.475** | **0.821** | **0.534** | 0.646 | **0.702** | **0.626** | **0.575** | **0.696** | 0.626 | **0.713** | **0.643** |

Figure 6: The curve of the reward values of RLF-KG training over three different datasets.

to generate a sequence of tokens based on an observation set. Since the order of information in the observation set is not important, the transformer encoder is more likely to learn the occurrence of items instead of the dependencies among entities for generating a hypothesis. Figure 6 serves as a complement to the previous statement by illustrating the increasing reward throughout the PPO process. We also refer readers to the Appendix G for qualitative examples demonstrating demonstrate the improvement in the generated hypotheses for the same observation.

## 4.5 ADDING STRUCTURAL REWARD TO PPO

In this part, we further analyze whether it is better to also include the structural similarity to the reward function used in PPO training. In the original setting of RLF-KG, we only include the Jaccard index between the given observation and the conclusion drawn from the training KG into the reward. It is also possible to incorporate another reward function that considers the structural differences between the generated and sampled hypotheses. As introduced before, the structural similarity can be measured by the SMATCH score. We also conducted further experiments to also include $S(H, H_{ref})$ as an additional term of the reward function, and the results are shown in Table 4. As SMATCH scores suggest, by incorporating the structural reward, the model can indeed generate hypotheses that are closer to the reference hypotheses. However, the Jaccard scores show that with structural information incorporated, the overall performance is comparable to or slightly worse than the original reward function.

## 4.6 COMPARISON BETWEEN SEARCH-BASED METHODS

In this section, we conduct a comparison of inference time and performance between the generation-based method and search-based methods. To do this comparison, we introduce a brute-force search algorithm. For each given observation, the algorithm, as detailed in Appendix C, explores all potential 1p hypotheses within the training graph and selects the one with the highest Jaccard similarity concerning the training graph. Table 5 shows that, notably, generation-based models of both architectures consistently exhibit significantly faster performance compared to the search-based method. Table 6 shows that, while our generation model only slightly overperforms the search-based method in Jaccard performance, it is significantly better in SMATCH performance. In many scenarios, SMATCH remains a critical metric to consider.

Table 4: The Jaccard and SMTACH performance of different reward functions.

| | FB15k-237 | | WN18RR | | DBpedia50 | |
|---|---|---|---|---|---|---|
| | Jaccard | SMATCH | Jaccard | SMATCH | Jaccard | SMATCH |
| Encoder-Decoder | 0.565 | **0.602** | 0.708 | **0.558** | 0.627 | 0.486 |
| + RLF-KG (Jaccard) | **0.666** | 0.530 | 0.753 | 0.540 | **0.731** | **0.541** |
| + RLF-KG (Jaccard + SMATCH) | 0.660 | 0.568 | **0.757** | 0.545 | 0.696 | 0.532 |
| Decoder-Only | 0.601 | **0.614** | 0.692 | **0.564** | 0.623 | **0.510** |
| + RLF-KG (Jaccard) | **0.660** | 0.598 | **0.726** | 0.518 | 0.643 | 0.492 |
| + RLF-KG (Jaccard + SMATCH) | 0.656 | 0.612 | 0.713 | 0.540 | **0.645** | 0.504 |

Table 5: Runtime for inference for various methods on testing data.

| Method | FB15k-237 | WN18RR | DBpedia50 |
|---|---|---|---|
| Brute-force Search | 11 days 8 hrs 25 mins | 2 days 20 hrs 4 mins | 18 hrs 52 mins |
| Generation + RLF-KG | **4 hrs 24 mins** | **32 mins** | **5 mins** |

Table 6: Performance of various methods on testing data.

| Method | FB15k-237 | | WN18RR | | DBpedia50 | |
|---|---|---|---|---|---|---|
| | Jaccard | SMATCH | Jaccard | SMATCH | Jaccard | SMATCH |
| Brute-force Search | 0.635 | 0.305 | 0.742 | 0.322 | 0.702 | 0.322 |
| Generation + RLF-KG | **0.666** | **0.530** | **0.753** | **0.540** | **0.731** | **0.541** |

## 5 RELATED WORK

The problem of abductive knowledge graph reasoning shares connections with various other knowledge graph reasoning tasks, including knowledge graph completion, complex logical query answering, and rule mining. Rule mining is line of work focusing on inductive logical reasoning, namely discovering logical rules over the knowledge graph. Various methods are proposed in this line of work (Galárraga et al., 2015; Ho et al., 2018; Meilicke et al., 2019; Cheng et al., 2022; 2023). In a different study, Dai et al. (2019) suggests using abductive learning (ABL) to create symbolic representations through learning methods, and then employing Prolog's abductive logic programming to solve hand-written puzzles. The resulting symbolic representations may not be logical expressions, and Prolog's abductive logic programming can only determine if they are true or false, without generating complex first-order structured hypotheses.

Complex logical query answering is a task of answering logically structured queries on KG. Query answering is a deduction process but involves knowledge induction on KG to generalize to unknown facts. Query embedding is a fast and robust method for complex query answering. Their primary focus is the enhancement of embedding structures for encoding sets of answers (Hamilton et al., 2018; Sun et al., 2020; Liu et al., 2021). For instance, Ren & Leskovec (2020) and Zhang et al. (2021b) introduce the utilization of geometric structures such as rectangles and cones within hyperspace to represent entities. Neural MLP (Mixer) (Amayuelas et al., 2022) use MLP and MLP-Mixer as the operators. Bai et al. (2022) suggests employing multiple vectors to encode queries, thereby addressing the diversity of answer entities. FuzzQE (Chen et al., 2022) uses fuzzy logic to represent logical operators. Probabilistic distributions can also serve as a means of query encoding (Choudhary et al., 2021a;b), with examples including Beta Embedding (Ren & Leskovec, 2020) and Gamma Embedding (Yang et al., 2022).

## 6 CONCLUSION

In summary, this paper has introduced the task of abductive logical knowledge graph reasoning. Meanwhile, this paper has proposed a generation-based method to address knowledge graph incompleteness and reasoning efficiency by generating logical hypotheses. Furthermore, this paper demonstrates the effectiveness of our proposed reinforcement learning from knowledge graphs (RLF-KG) to enhance our hypothesis generation model by leveraging feedback from knowledge graphs.

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

## A  ALGORITHM FOR SAMPLING OBSERVATION-HYPOTHESIS PAIRS

In this section, we present the Algorithm 1 used to sample complex hypotheses from a given knowledge graph. Given a knowledge graph $\mathcal{G}$ and a hypothesis type $T$, the algorithm starts with a random node $v$ and recursively constructs a hypothesis that has $v$ in its conclusions and the corresponding structure $T$. During the recursion process, the algorithm examines the last operation in the current hypothesis. If the operation is *projection*, the algorithm randomly selects one of its predecessors $u$ that holds the corresponding relation to $v$ as the answer of its sub-hypothesis. The algorithm then calls the recursion on node $u$ and the sub-hypothesis type of $T$ again. Similarly, for *intersection* and *union*, the algorithm applies recursion on their sub-hypothesis on the same node $v$. The recursion stops when the current node contains an entity.

---

**Algorithm 1** Sampling Hypothesis According to Type

---
    **Input** Knowledge graph $\mathcal{G}$, hypothesis type $T$
    **Output** Hypothesis sample
    **procedure** SAMPLEHYPOTHESIS($\mathcal{G}, T$)
        **function** GROUNDTYPE($\mathcal{G}, T, tail$)
            **if** $T.operation = p$ **then**
                $head \leftarrow$ SAMPLE($\{head | (head, tail) \text{is an edge in } \mathcal{G}\}$)
                $RelType \leftarrow$ type of $(head, tail)$ in $\mathcal{G}$
                $ProjectionType \leftarrow p$
                $child \leftarrow$ the only child in $T.children$
                $SubHypothesis \leftarrow$ GROUNDTYPE($\mathcal{G}, child, head$)
                **return** $(ProjectionType, RelType, SubHypothesis)$
            **else if** $T.operation = i$ **then**
                $IntersectionResult \leftarrow (i)$
                **for** $child \in T.children$ **do**
                    $SubHypothesis \leftarrow$ GROUNDTYPE($\mathcal{G}, child, tail$)
                    $IntersectionResult.$PUSHBACK($child, tail$)
                **end for**
                **return** $IntersectionResult$
            **else if** $T.operation = u$ **then**
                $UnionResult \leftarrow (u)$
                **for** $child \in T.children$ **do**
                    **if** $child$ is the first child **then**      ▷ For the first subquery, we choose the current root node.
                      $tail \leftarrow$ SAMPLE($\mathcal{G}$)        ▷ For other subquery, the root node can be any vertex
                  **end if**
                  $SubHypothesis \leftarrow$ GROUNDTYPE($\mathcal{G}, T.child, tail$)
                  $UnionResult.$PUSHBACK($child, tail$)
                **end for**
                **return** $UnionResult$
            **else if** $T.operation = e$ **then**
                **return** $(e, T.value)$
            **end if**
        **end function**
        $v \leftarrow$ an arbitrary vertex in $\mathcal{G}$
        **return** GROUNDTYPE($\mathcal{G}, T, v$)
    **end procedure**

---

## B  ALGORITHMS FOR CONVERSION BETWEEN QUERIES AND ACTIONS

In this part, we present the details of tokenizing the hypothesis graph (Algorithm 2), and formulating a graph according to the tokens, namely the process of de-tokenization (Algorithm 3). In this part, we also called the tokens "actions" because the algorithms are inspired by the action-based semantic parsing algorithms. Note that we use the notations $G, V, E$ for the hypothesis graph to distinguish it from the knowledge graph.

---

**Algorithm 2** HypothesisToActions

---

    **Input** Hypothesis plan graph $G$
    **Output** Action sequence $A$
  **procedure** HYPOTHESISTOACTIONS($G$)
      **function** DFS($G, tail, A$)
          **if** $tail$ is an anchor node **then**
              $action \leftarrow$ the entity $tail$ represents
          **else**
              $action \leftarrow$ the operator the first in-edge of $tail$ represents
          **end if**
          Append $action$ to $A$
          **for all** in-edges to $tail$ in $G$ $(head, rel, tail)$ **do**
              DFS($G, head, A$)
          **end for**
      **end function**
      $root \leftarrow$ the root of $G$
      $A \leftarrow$ DFS($G, root, \emptyset$)
      **return** $A$
  **end procedure**

---

**Algorithm 3** ActionsToHypothesis

---

    **Input** Action sequence $A$
    **Output** Hypothesis plan graph $G$
  **procedure** ACTIONSTOHYPOTHESIS($A$)
      $S \leftarrow$ an empty stack
      Create an map $deg$, $deg[i] = deg[u] = 2$ and 1 for other operators.
      $V \leftarrow \emptyset, E \leftarrow \emptyset$
      **for** $action \in A$ **do**
          Create a new node, denoted by $head$, $V \leftarrow V \cup \{head\}$
          **if** $S \neq \emptyset$ **then**
              $(tail, operator, degree) \leftarrow$ the top element in $S$
              $E \leftarrow E \cup \{(head, operator, tail)\}$
          **end if**
          **if** $action$ represents an anchor node **then**
              Mark $head$ as an anchor node with entity $action$
              **while** $S \neq \emptyset$ **do**
                 Pop the top element $(tail, operator, degree)$ from $S$
                 $degree \leftarrow degree - 1$
                 **if** $degree > 0$ **then**
                    Push $(tail, operator, degree)$ to $S$
                    Break
                 **end if**
              **end while**
          **else**
              Push $(head, action, deg[action])$ to $S$
          **end if**
      **end for**
      $G \leftarrow (V, E)$
      **return** $G$
  **end procedure**

---

## C  ALGORITHMS FOR ONE-HOP SEARCHING

In this part, we demonstrate the algorithm used for searching the best relation-tail pairs among all entities in the observation set as the one-hop hypothesis to explain the observations.

---

**Algorithm 4** One-Hop-Search

---

    **Input** Observation set $O$
    **Output** Hypothesis $bestHypothesis$
  **procedure** ONE-HOP-SEARCH($O$)
      $candidates \leftarrow \{(h, r, t) \in \mathcal{R}_{\text{train}} | t \in O\}$
      $bestJaccard \leftarrow 0, bestHypothesis \leftarrow$ Null
      **for** $(h, r, t) \in candidates$ **do**
          $H \leftarrow$ the one-hop hypothesis formed by the single edge $(h, r, t)$
          $trainJaccard \leftarrow$ Jaccard($[\![H]\!]_{\mathcal{G}_{\text{train}}}, A$)
          **if** $trainJaccard > bestJaccard$ **then**
              $bestJaccard \leftarrow trainJaccard$
              $bestHypothesis \leftarrow H$
          **end if**
      **end for**
      **return** $bestHypothesis$
  **end procedure**

---

## D  DETAILS OF USING SMATCH TO EVALUATE STRUCTURAL DIFFERNECES OF QUERIES

Smatch Cai & Knight (2013) is an evaluation metric for Abstract Meaning Representation (AMR) graphs. An AMR graph is a directed acyclic graph with two types of nodes: variable nodes and concept nodes, and three types of edges:

- Instance edges, which connect a variable node to a concept node and are labeled literally "instance". Every variable node must have exactly one instance edge, and vice versa.

- Attribute edges, which connect a variable node to a concept node and are labeled with attribute names.

- Relation edges, which connect a variable node to another variable node and are labeled with relation names.

Given a predicted AMR graph $G_{pred}$ and the gold AMR graph $G_{gold}$, the Smatch score of $G_{pred}$ with respect to $G_{gold}$ is denoted by SMATCH($G_{pred}, G_{gold}$). SMATCH($G_{pred}, G_{gold}$) is obtained by finding an approximately optimal mapping between the variable nodes of the two graphs and then matching the edges of the graphs.

Our hypothesis graph is similar to the AMR graph, in:

- The nodes are both categorized as fixed nodes and variable nodes

- The edges can be categorized into two types: edges from a variable node to a fixed node and edges from a variable node to another variable node. And edges are labeled with names.

However, they are different in that the AMR graph requires every variable node to have instance edges, while the hypothesis graph does not.

The workaround for leveraging the Smatch score to measure the similarity between hypothesis graphs is creating an instance edge from every entity to some virtual node. Formally, given a hypothesis $H$ with hypothesis graph $G(H)$, we create a new hypothesis graph $G_A(H)$ to accommodate the AMR settings as follows: First, we initialize $G_A(H) = G(H)$. Then, create a new relation type $instance$ and add a virtual node $v'$ into $G_A(H)$. Finally, for every variable node $v \in G(H)$, we add a relation $instance(v, v')$ into $G_A(H)$. Then, given a predicted hypothesis $H_{pred}$ and a gold

hypothesis $H_{gold}$, the Smach score is defined as

$$S(H_{pred}, H_{gold}) = \texttt{SMATCH}(G_A(H_{pred}), G_A(H_{gold})). \tag{9}$$

Through this conversion, a variable entity $v_g$ of $H_{gold}$ is mapped to a variable entity $v_p$ of $H_{pred}$ if and only if $instance(v_g, v')$ is matched with $instance(v_p, v')$. This modification does not affect the overall algorithm for finding the optimal mapping between variable nodes and hence gives a valid and consistent similarity score. However, this adds an extra point for matching between instance edges, no matter how the variable nodes are mapped.

# E    DETAILED SCORES BY QUERY TYPES

In this part, we show the detailed SMATCH performance of various methods.

Table 7: The detailed SMATCH performance of various methods.

| Dataset | Model | 1p | 2p | 2i | 3i | ip | pi | 2u | up | 2in | 3in | pni | pin | inp | Ave. |
|---|---|---|---|---|---|---|---|---|---|---|---|---|---|---|---|
| FB15k-237 | Enc.-Dec. | 0.342 | 0.506 | 0.595 | 0.602 | 0.570 | 0.598 | 0.850 | 0.571 | 0.652 | 0.641 | 0.650 | 0.655 | 0.599 | 0.602 |
| | RLF-KG (J) | 0.721 | 0.643 | 0.562 | 0.480 | 0.364 | 0.475 | 0.769 | 0.431 | 0.543 | 0.499 | 0.513 | 0.518 | 0.370 | 0.530 |
| | RLF-KG (J+S) | 0.591 | 0.583 | 0.577 | 0.531 | 0.447 | 0.520 | 0.820 | 0.505 | 0.602 | 0.563 | 0.571 | 0.595 | 0.484 | 0.568 |
| | Dec.-Only | 0.287 | 0.481 | 0.615 | 0.623 | 0.599 | 0.626 | 0.847 | 0.574 | 0.680 | 0.656 | 0.675 | 0.677 | 0.636 | 0.614 |
| | RLF-KG (J) | 0.344 | 0.445 | 0.675 | 0.585 | 0.537 | 0.638 | 0.853 | 0.512 | 0.696 | 0.575 | 0.647 | 0.688 | 0.574 | 0.598 |
| | RLF-KG (J+S) | 0.303 | 0.380 | 0.692 | 0.607 | 0.565 | 0.671 | 0.857 | 0.506 | 0.727 | 0.600 | 0.676 | 0.734 | 0.634 | 0.612 |
| WN18RR | Enc.-Dec. | 0.375 | 0.452 | 0.591 | 0.555 | 0.437 | 0.585 | 0.835 | 0.685 | 0.586 | 0.516 | 0.561 | 0.549 | 0.530 | 0.558 |
| | RLF-KG (J) | 0.455 | 0.468 | 0.563 | 0.562 | 0.361 | 0.530 | 0.810 | 0.646 | 0.560 | 0.530 | 0.536 | 0.539 | 0.465 | 0.540 |
| | RLF-KG (J+S) | 0.443 | 0.457 | 0.565 | 0.572 | 0.366 | 0.545 | 0.814 | 0.661 | 0.541 | 0.553 | 0.532 | 0.546 | 0.491 | 0.545 |
| | Dec.-Only | 0.320 | 0.443 | 0.582 | 0.551 | 0.486 | 0.597 | 0.809 | 0.696 | 0.594 | 0.526 | 0.575 | 0.574 | 0.577 | 0.564 |
| | RLF-KG (J) | 0.400 | 0.438 | 0.566 | 0.491 | 0.403 | 0.519 | 0.839 | 0.667 | 0.547 | 0.450 | 0.497 | 0.466 | 0.450 | 0.518 |
| | RLF-KG (J+S) | 0.375 | 0.447 | 0.584 | 0.499 | 0.432 | 0.545 | 0.825 | 0.679 | 0.584 | 0.477 | 0.543 | 0.522 | 0.507 | 0.540 |
| DBpedia50 | Enc.-Dec. | 0.345 | 0.396 | 0.570 | 0.548 | 0.344 | 0.576 | 0.712 | 0.544 | 0.474 | 0.422 | 0.477 | 0.488 | 0.428 | 0.486 |
| | RLF-KG (J) | 0.461 | 0.424 | 0.634 | 0.584 | 0.361 | 0.575 | 0.809 | 0.579 | 0.584 | 0.497 | 0.544 | 0.533 | 0.454 | 0.541 |
| | RLF-KG (J+S) | 0.419 | 0.410 | 0.638 | 0.555 | 0.373 | 0.586 | 0.785 | 0.579 | 0.560 | 0.459 | 0.536 | 0.542 | 0.474 | 0.532 |
| | Dec.-Only | 0.378 | 0.408 | 0.559 | 0.526 | 0.397 | 0.568 | 0.812 | 0.626 | 0.480 | 0.414 | 0.489 | 0.494 | 0.474 | 0.510 |
| | RLF-KG (J) | 0.405 | 0.411 | 0.558 | 0.496 | 0.376 | 0.507 | 0.825 | 0.621 | 0.477 | 0.397 | 0.468 | 0.444 | 0.406 | 0.492 |
| | RLF-KG (J+S) | 0.398 | 0.415 | 0.567 | 0.497 | 0.383 | 0.533 | 0.827 | 0.630 | 0.510 | 0.420 | 0.484 | 0.457 | 0.430 | 0.504 |

Table 8: The detailed SMATCH performance of the searching method.

| Dataset | 1p | 2p | 2i | 3i | ip | pi | 2u | up | 2in | 3in | pni | pin | inp | Ave. |
|---|---|---|---|---|---|---|---|---|---|---|---|---|---|---|
| FB15k-237 | 0.945 | 0.340 | 0.365 | 0.218 | 0.184 | 0.267 | 0.419 | 0.185 | 0.301 | 0.182 | 0.245 | 0.155 | 0.157 | 0.305 |
| WN18RR | 0.957 | 0.336 | 0.420 | 0.274 | 0.182 | 0.275 | 0.427 | 0.183 | 0.323 | 0.224 | 0.270 | 0.155 | 0.156 | 0.322 |
| DBpedia | 0.991 | 0.336 | 0.399 | 0.259 | 0.182 | 0.245 | 0.441 | 0.183 | 0.332 | 0.226 | 0.290 | 0.154 | 0.155 | 0.322 |

# F    HYPERPARAMETERS OF THE RL PROCESS

The PPO hyperparameters are shown as in Table 9. We warm-uped the learning rate from $0.1$ of the peak to the peak value within the first $10\%$ of total iterations and then decayed it to $0.1$ of the peak with a cosine schedule.

Table 9: PPO Hyperparamters.

| Hyperparam. | Enc.-Dec. | | | Dec.-Only | | |
|---|---|---|---|---|---|---|
| | FB15k-237 | WN18RR | DBpedia50 | FB15k-237 | WN18RR | DBpedia50 |
| Learning rate | 2.4e-5 | 3.1e-5 | 1.8e-5 | 0.8e-5 | 0.8e-5 | 0.6e-5 |
| Batch size | 16384 | 16384 | 4096 | 3072 | 2048 | 2048 |
| Minibatch size | 512 | 512 | 64 | 96 | 128 | 128 |
| Horizon | 4096 | 4096 | 4096 | 2048 | 2048 | 2048 |

Table 10: Example FB15k-237 Case study 1.

| | | |
|---|---|---|
| **Sample** | Interpretation | Companies operating in industries that intersect with Yahoo! but not with IBM. |
| | Hypothesis | The observations are the $V_?$ such that $\exists V_1, inIndustry(V_1, V_?) \wedge \neg industryOf(IBM, V_1) \wedge industryOf(Yahoo!, V_1)$ |
| | Observation | EMI, Columbia, Viacom, Yahoo!, Bandai, Rank_Organisation, Gannett_Company, NBCUniversal, Pony_Canyon, The_Graham_Holdings_Company, Televisa, Google, Microsoft_Corporation, Munhwa_Broadcasting_Corporation, CBS_Corporation, GMA_Network, Victor_Entertainment, Sony_Music_Entertainment_(Japan)_Inc., Toho_Co.,_Ltd., The_New_York_Times_Company, Star_Cinema, TV5, Avex_Trax, The_Walt_Disney_Company, Metro-Goldwyn-Mayer, Time_Warner, Dell, News_Corporation |
| **Searching** | Interpretation | Which companies operate in media industry? |
| | Hypothesis | The observations are the $V_?$ such that $inIndustry(Media, V_?)$ |
| | Conclusion | Absent: - Google, - Microsoft_Corporation, - Dell |
| | Jaccard | 0.893 |
| | Smatch | 0.154 |
| **Enc.-Dec.** | Interpretation | Companies operating in industries that intersect with Yahoo! but not with Microsoft Corporation. |
| | Hypothesis | The observations are the $V_?$ such that $\exists V_1, inIndustry(V_1, V_?) \wedge \neg industryOf(Microsoft\_Corporation, V_1) \wedge industryOf(Yahoo!, V_1)$ |
| | Conclusion | Absent: Microsoft_Corporation |
| | Jaccard | 0.964 |
| | Smatch | 0.909 |
| **+ RLF-KG** | Interpretation | Companies operating in industries that intersect with Yahoo! but not with Oracle Corporation. |
| | Hypothesis | The observations are the $V_?$ such that $\exists V_1, inIndustry(V_1, V_?) \wedge \neg industryOf(Oracle\_Corporation, V_1) \wedge industryOf(Yahoo!, V_1)$ |
| | Concl. | Same |
| | Jaccard | 1.000 |
| | Smatch | 0.909 |

# G  CASE STUDIES

In this session, we show some concrete examples in Table 10, 11 and 12 that are given by different abductive reasoning methods, namely searching, generation model with supervised training, and generation model with supervised training with RLF-KG.

Table 11: FB15k-237 Case study 2.

| | | |
|---|---|---|
| **Sample** | Interpretation | Locations that adjoin second-level divisions of the United States of America that adjoin Washtenaw County. |
| | Hypothesis | The observations are the $V_?$ such that $\exists V_1, adjoins(V_1, V_?) \wedge adjoins(Washtenaw\_County, V_1) \wedge secondLevelDivisions(USA, V_1)$ |
| | Observation | Jackson_County, Macomb_County, Wayne_County, Ingham_County Washtenaw_County, |
| **Searching** | Interpretation | Locations that adjoin Oakland County. |
| | Hypothesis | The observations are the $V_?$ such that $adjoins(Oakland\_County, V_?)$ |
| | Conclusion | Absent: - Jackson_County - Ingham_County |
| | Jaccard | 0.600 |
| | Smatch | 0.182 |
| **Enc.-Dec.** | Interpretation | Second-level divisions of the United States of America that adjoin locations that adjoin Oakland County. |
| | Hypothesis | The observations are the $V_?$ such that $\exists V_1, secondLevelDivisions(USA, V_?) \wedge adjoins(V_1, V_?) \wedge +adjoins(Oakland\_County, V_1)$ |
| | Conclusion | Extra: Oakland_County Absent: Wayne_County |
| | Jaccard | 0.667 |
| | Smatch | 0.778 |
| **+ RLF-KG** | Interpretation | Second-level divisions of the United States of America that adjoin locations contained within Michigan. |
| | Hypothesis | The observations are the $V_?$ such that $\exists V_1, secondLevelDivisions(USA, V_?) \wedge adjoins(V_1, V_?) \wedge containedIn(Michigan, V_1)$ |
| | Conclusion | Extra: - Oakland_County - Genesee_County |
| | Jaccard | 0.714 |
| | Smatch | 0.778 |

Table 12: DBpedia50 Case study.

| | | |
|---|---|---|
| **Ground Truth** | Interpretation | Works, except for "Here 'Tis," that have subsequent works in the jazz genre. |
| | Hypothesis | The observations are the $V_?$ such that $\exists V_1, subsequentWork(V_1, V_?) \wedge \neg previousWork(Here\_'Tis, V_1) \wedge genre(Jazz, V_1)$ |
| | Observation | Deep\_Deep\_Trouble, Good\_Dog,\_Happy\_Man, I\_Don't\_Want\_to\_Be\_Your\_Friend, Interior\_Music,     Lee\_Morgan\_Sextet, Paris\_Nights\\/New\_York\_Mornings, Take\_the\_Box |
| **Searching** | Interpretation | Works subsequent to "Closer" (Corinne Bailey Rae song). |
| | Hypothesis | The observations are the $V_?$ such that $subsequentWork(Closer\_(Corinne\_Bailey\_Rae\_song), V_?)$ |
| | Conclusion | Only Paris\_Nights\\/New\_York\_Mornings |
| | Jaccard | 0.143 |
| | Smatch | 0.154 |
| **Enc.-Dec.** | Interpretation | Works, except for "Lee Morgan Sextet," that have subsequent works in the jazz genre. |
| | Hypothesis | The observations are the $V_?$ such that $\exists V_1, subsequentWork(V_1, V_?) \wedge \neg previousWork(Lee\_Morgan\_Sextet, V_1) \wedge genre(Jazz, V_1)$ |
| | Conclusion | Extra: Here\_'Tis Absent: Lee\_Morgan\_Sextet |
| | Jaccard | 0.750 |
| | Smatch | 0.909 |
| **+ RLF-KG** | Interpretation | Works that have subsequent works in the jazz genre. |
| | Hypothesis | The observations are the $V_?$ such that $\exists V_1, subsequentWork(V_1, V_?) \wedge genre(Jazz, V_1)$ |
| | Conclusion | Extra: Here\_'Tis |
| | Jaccard | 0.875 |
| | Smatch | 0.400 |

