# OpenReview forum: "Abductive Logical Reasoning on Knowledge Graphs"
_ICLR.cc/2024/Conference — Submitted to ICLR 2024_

### Official Review · Reviewer_xMZe · 2023-10-29

**Soundness:** 2 fair
**Presentation:** 2 fair
**Contribution:** 3 good
**Rating:** 3
**Confidence:** 4

**Summary:**

This paper introduces the task of abductive logical knowledge graph reasoning. It proposes a generation-based method to address knowledge graph incompleteness and reasoning efficiency by generating logical hypotheses. The paper demonstrates the effectiveness of their proposed reinforcement learning from knowledge graphs (RLF-KG) to enhance the hypothesis generation model by leveraging feedback from knowledge graphs. The paper also discusses the tokenization of hypotheses and the use of reinforcement learning to optimize the hypothesis generation model. Overall, the paper aims to utilize information from knowledge graphs to find complex structured hypotheses that explain observations.

**Strengths:**

- The research shows that the generation-based method consistently outperforms the search-based method on three different datasets.
- The method addresses the complexity of logical hypotheses by using a generation-based approach. This allows for the exploration of complex structured hypotheses that can explain the observations, going beyond simple correlations or patterns.
- The method takes into account the incompleteness of the knowledge graph, which is a common challenge in reasoning tasks. By generating logical hypotheses, the method can fill in the gaps and provide explanations for the given observations.

**Weaknesses:**

- It is inappropriate to call this task as "abduction". The task learns a first-order hypothesis that satisfies given groundings, and the generated hypothesis is used for generalising to more unseen groundings. Logical abduction means a grounding-to-grounding hypothesizing, for example, given P(x)->Q(x), observing Q(1), we can abduce P(1). If you want to abduce first-order theories, the you need second-order rules as background knowledge.
- The presentation of this work can be significantly improved. The description of the proposed method is messy. For example, the caption of Figure 4 is "step 3", which is out of blue and makes reader confusing. Furthermore, the illustration of figures and algorithms are also confusing. There's no input and output in algorithms, and what are "models" and "reference models" in Fig. 4?
- Fig. 5 proposes 13 types of first-order hypotheses templates, are they complete for the hypothesis space? Is there any proof to the completeness?

**Questions:**

Please see my above comments.

---

> ### Author Response · Authors · 2023-11-15
>
> Thank you for taking the time to review our work. We appreciate your feedback and would like to respond to your comments and questions.
>
> ## Re W1:
>
> In our formulation of abductive reasoning on knowledge graphs, given a grounded observation such as Q(1), the goal is the find a hypothesis P(x) such that the set {x | P(x) holds true} closely resembles the set 1. This means that P(1) = True.
>
> To provide a clearer understanding, we will compare our abductive logical expression generation with inductive rule mining on knowledge graphs:
>
> Rule Mining (ILP on KG):
>
> This task involves a given knowledge graph with some positive examples regarding the relation $speak$. The goal is to find Horn clause rules associated with the $speak$ relation.
>
> $speaks(X, Y) \\leftarrow lives(X, A) \\land language(A, Y)$
>
> The variables in the rules are universally quantified, and the quality of the rules is measured by head coverage and precision.
>
> Our task:
>
> We use a real example from FB15K-237. Our observations include the following entities:
>
> $Grant Heslov, Jason Segel, Robert Towne, Ronald Bass, Rashida Jones$
>
> Our task aims to find a logical expression based on the KG to describe these entities, rather than discovering general rules over the KG. These entities make the following logical expressions true:
>
> $V_?: Occupation(V_?, Actor) \\land Occupation(V_?, Screen Writer) \\land BornIn(V_?, Los Angeles)$
>
> This means that Grant Heslov, Jason Segel, Robert Towne, Ronald Bass, and Rashida Jones are actors and screenwriters born in Los Angeles. This logical expression is not a rule, as not all entities from the KG satisfy the logical expression.
>
> Unlike inductive rule mining, which is not grounded, our proposed reasoning task is grounded, focusing on specific entities.
>
> ## Re W2:
>
> We have updated the description of our proposed method, RLF-KG, in Section 3 to improve clarity. The entire process is divided into three steps.
>
> Step 1 involves preparing the training samples and converting the raw data into a format that is suitable for a transformer-based model.
>
> In Step 2, a generative model is trained in a supervised manner. It is important to note that the resulting model from Step 2 is standalone and can be used to generate hypotheses given observations, although its performance may be relatively low.
>
> The primary contribution of our method is Step 3, which fine-tunes the trained model via PPO with feedback from the training KG. This step is critical in improving the model's ability to reason over the KG and generate more accurate and complex hypotheses.
> In regard to the "model" and "reference model," both are initialized to be the trained model resulting from Step 2 (which we refer to as the "original model"). During Step 3, the "model" is optimized, while the "reference model" remains fixed. The "reference model" is used in the PPO process to prevent excessive divergence between the fine-tuned "model" and the "original model."
> The "reference model" is crucial in ensuring that the fine-tuned "model" produces syntactically correct hypotheses similar to the "original model." Without limiting divergence, the training process may become difficult to converge.
> Our quantitative experiments (presented in Table 3, Sec. 4.4) demonstrated the effectiveness of Step 3, namely RLF-KG. Figure 6 illustrates the growth in reward during the PPO process, further supporting the effectiveness of our proposed method.
>
> In addition to the quantitative results, we provide qualitative examples in Appendix G to showcase the improvement in the generated hypotheses for the same observation. These examples illustrate how RLF-KG improves the model's ability to reason over the KG and generate more accurate and complex hypotheses.
>
> ## Re W3:
>
> In line with previous works on KG reasoning [1][2], we utilize similar hypothesis templates as those used in these studies. Due to the exponential increase in the number of logical expression templates with the number of variables, we are only able to evaluate a subset of them in our experiments. However, the selected expression templates are considered empirically significant based on their use in various previous works on complex query answering [1][2].
>
> [1] Query2box: Reasoning over knowledge graphs in vector space using box embeddings.
>
> [2] Beta embeddings for multi-hop logical reasoning in knowledge graphs.

---

> > ### Comment · Reviewer_xMZe · 2023-11-18
> >
> > I thank the authors for their clarifications in __Re W2__, however, the example you gave in __Re W1__ is an induction, since it learns a first-order rule that has a set of answers. And the empirical results in __Re W3__ cannot guarantee completeness, which think is fairly easy to prove, for example:
> > - A. Cropper and S.H. Muggleton. Logical minimisation of meta-rules within meta-interpretive learning. In Proceedings of the 24th International Conference on Inductive Logic Programming, pages 65-78. Springer-Verlag, 2015. LNAI 9046.
> >
> > Therefore, I think this work needs significant improvement before published, so I will keep my rating.

---

> > ### Author Response · Authors · 2023-11-19
> >
> > Thank you for your response. Allow us to provide additional clarifications:
> >
> > Re W1:
> > - We would like to respectfully discuss the assertion that "the generated hypothesis is used for generalizing to more unseen groundings" in W1. Our intention is **not** to have the generated hypotheses **generalize to unseen groundings**, but rather to **exclusively** serve as the **best explanation** for the given observation. This objective is reflected in our experimental metric, where we gauge the Jaccard similarity between the set of entities making the generated hypothesis true and the given observation. In fact, the **Jaccard similarity decreases** as the generated hypothesis generalizes to **more unseen groundings**, as the overlap between the two sets accounts for a smaller fraction of their union.
> > - Quoting from [1], “It suggests that the best way to distinguish between induction and abduction is this: both are ampliative, meaning that the conclusion goes beyond what is (logically) contained in the premises …, but in **abduction** there is an implicit or explicit **appeal to explanatory considerations**, whereas in induction there is not; in induction, there is only an appeal to observed frequencies or statistics.”. Our goal, in line with the description of abduction provided here, is to generate an explanatory hypothesis for a given observation, **not** a rule intended to hold **generally** with high probability.
> > - Returning to our example, the hypothesis presented is just a sample. Our expectation is not for it to be a rule but, ideally, to hold true only for the specified five celebrities. Considering the possibility of additional groundings in reality, the hypothesis may not be the best explanation, and it's essential to identify exclusive common characteristics among the five celebrities for a more refined hypothesis.
> >
> > Re W2:
> > - On one hand, we have not asserted that our scope is exhaustive. As stated in the paper, the number of candidate hypotheses is **combinatorial** concerning the nodes and relations in a hypothesis type, making it computationally expensive to consider an extensive range of hypothesis types.
> > - On the other hand, we **do not restrict** our generative model to producing only the 13 types of hypotheses, as mentioned in Sec. 4.1. The 13 types are employed in the **sampling process**. As long as the generated hypotheses are **syntactically correct**, our program is able to find the corresponding conclusion, compare it with the given observation, and **produce a reward** for the PPO. This also explains why we developed the PPO method on top of supervised learning, as the supervision signal only compares the sampled hypotheses (of one of the 13 types) with the generated hypotheses.
> >
> > [1]  Douven, Igor, "Abduction", The Stanford Encyclopedia of Philosophy (Summer 2021 Edition), Edward N. Zalta (ed.), URL = <https://plato.stanford.edu/archives/sum2021/entries/abduction/>.

---

> > > ### Comment · Reviewer_xMZe · 2023-11-19
> > >
> > > Thanks again for the reply. The first-order rules induced from a set of groundings can be regarded as abduction, as long as you use second-order rules as background knowledge, like your templates in the paper. It is a bit vague to discriminate induction,
> > > abduction and even deduction  philosophically, actually all of them can be implemented by deduction (i.e. resolution)
> > > in higher order logic, therefore it would be better to constrain the discussion within the context of mathematical logic, especially predicate logic.

---

### Official Review · Reviewer_jBpD · 2023-11-04

**Soundness:** 3 good
**Presentation:** 4 excellent
**Contribution:** 3 good
**Rating:** 6
**Confidence:** 3

**Summary:**

This paper explores an important problem of abductive logical reasoning  (ALR) over Knowledge Graphs (KG), discusses a practical approach to implementing ALR over KGs, and compares different possible implementations of the proposed approach to each other.  ALR over KG is defined as inferring the most probable logic hypothesis from the KGs to explain an observed entity set, and a generative approach is presented for creating logical expressions based on entity set observations.  Algorithms are presented for (1) sampling hypothesis-observation pairs from a KG under the open-world assumption and (2) implementing reinforcement training from KG feedback.  Additional transformation algorithms (token-to-graph and graph-to-token) that assist in the sampling process are presented.  The authors demonstrate the relative effectiveness of each sub-component of their overall method on three well know datasets.

**Strengths:**

Originality: The authors take on an important problem in knowledge graph reasoning, namely the abduction of plausible rules that could result in an observation set if applied to the KG.  A reasonable baseline algorithm is described and implemented for providing such a capability.

Quality:  All aspects of the paper appear to be technically correct, and variations of the method are experimentally explored across three KGs.  Authors documentation on all aspects of the techniques are sufficient for reproducibility.

Clarity:  The main effort of the work exceptionally well described, providing good motivation, clear examples of what is being sought, how the algorithms work, and the various experiments performed.

Significance:  This is exploration is a valuable contribution to the literature.

**Weaknesses:**

Originality: The paper could benefit from a more precise statement of their objective problem, which would then allow for a clearer discussion of how the current effort contrasts with past and future work.

Quality: The objective of the approach is not precisely defined, so it is difficult to discern whether or not any particular intent was achieved.  No future research was discussed.

Clarity: covered by the first two items in this section.

Significance:  I feel that this is valuable, but could be more impactfully significant with a more precise framing of this approach and alternate formulations of abduction.

**Questions:**

1.  The paper criticizes previous approaches without quantitatively comparing them to the current method.  While this may be computationally infeasible on the graphs chosen here, do you have such results you could include -- even if only on very small graphs that highlight and justify your statements?

2. There are many possible formal definitions for abduction, and you choose two slightly different versions in your discussion (one in the abstract, and one in Section 2).  Can you provide a precise statement of what you mean by abduction, rather than proceed by analogy?  For instance, the last paragraph of section 2 mentions "best explanation" without providing a quotative definition of best.   Can you connect such precision to your actual implementation?  For instance a precise definition might be made to which your algorithm is an approximation, or a precise definition might be supplied for which your algorithm is an exact solution.

---

> ### Author Response · Authors · 2023-11-15
>
> Thank you for your review! We would like to make the following clarifications.
>
> ## Re W1:
>
> We revised and clarified the problem definition in section 2, which involves using abductive reasoning to find a hypothesis (H) that matches an observation set $O = \\{v_1, v_2, ..., v_k\\}$ as closely as possible. The objective of abductive reasoning is formalized in equation (4), which uses the Jaccard score to measure the similarity between the observations and the corresponding conclusions drawn from the generated conclusions on the hidden graph.
>
>
> ## Re W2:
>
> Please refer to Re:Q1 and Re:Q2 for the objectives of the problem and the approach. In addition, we further discuss the future work on these directions. One possible direction is to explore the use of more advanced reinforcement learning algorithms to improve the performance of the model. Another direction is to investigate the use of larger and more complex knowledge graphs. Our current experiments were conducted on relatively small graphs with 10k up to 100k vertices. It would be interesting to see how the model performs on larger graphs, as well as graphs with more complex structures and relationships.
>
>
> ## Re W4:
>
> We have revised the description of our approach in Section 3.2 to provide further clarity. Please refer to Re: Q2 for details.
>
> ## Re Q1:
>
> Our proposed KG reasoning task is unique and novel, which means there are no previous methods capable of generating complex first-order logical expressions to explain specific observations. In Table 5 and 6, we compare our method with pure symbolic methods, such as brute-force searching the one-hop neighbors of the observations. However, brute-force searching is very slow on graphs with at least 10k vertices, and it struggles to perform well when the hypothesis is complex and structured, involving multiple logical connectives and variables.
>
> In Appendix G Case Studies, we provide qualitative examples that compare the hypotheses produced by the search-based method, the generation-based method, and the generation-based method after applying RLF-KG.
>
> ## Re Q2:
>
> We want to clarify that the abductive reasoning objectives mentioned in the abstract and at the end of Section 2 are the same. We formally define the "best" explanation using formula (4) at the end of Section 2, which is represented by the Jaccard similarity between the conclusion of the generated hypothesis on the testing graph (which was not observed during training) and the observation.
>
> It is crucial to note that the reward function for the PPO is chosen to approximate the objective by using the Jaccard similarity between the conclusion of the generated hypothesis on the training graph (the observed graph) and the observation. Furthermore, the primary metric used in our experiments is consistent with the task's objective.

---

> > ### Comment · Reviewer_jBpD · 2023-11-18
> >
> > I thank the authors for their clarifications.  I believe this work represents an important step in addressing abductive reasoning, and that that the authors and other research groups can build upon in the future.  I will keep the Rating of 6.

---

### Official Review · Reviewer_dX8J · 2023-11-08

**Soundness:** 3 good
**Presentation:** 3 good
**Contribution:** 2 fair
**Rating:** 6
**Confidence:** 3

**Summary:**

This paper focuses on abductive reasoning with KGs and proposes a generative approach using supervised training to create logical expressions based on observations.   It further improves explanations by minimizing differences between observations and conclusions.      Experimental results show transformer-based models generate robust and efficient logical explanations, achieving state-of-the-art results on abductive reasoning with KGs.

**Strengths:**

This article provides an analysis of the difficulties associated with abductive reasoning in knowledge graphs (KGs) and proposes a generation-based approach to address these challenges. The overall structure of the article is well-organized, and the ideas are presented clearly. Additionally, the experimental findings are objective and cover a wide range of aspects.

**Weaknesses:**

1. This article lacks a clear formalization of the explanation for abductive reasoning.
2. The motivation for using reinforcement learning algorithms is questionable.
3. See Questions.

**Questions:**

1. Are the Encoder-Decoder and Decoder-only models used in the experimental section both based on the fundamental transformer architecture?    Does the author consider the substitution of more powerful backbone models?
2. The effectiveness of the PPO algorithm is not experimentally demonstrated in this paper.     In the context of the "abductive reasoning for KG" task, please explain why the PPO algorithm was chosen over other alternatives or how it specifically provides advantages.
3.  What are the reference models in Fig. 4? Why the method needs it?

---

> ### Author Response · Authors · 2023-11-15
>
> Thank you for your review, we would like to make the following clarifications on your comments.
>
> Re W1:
>
> We revised and clarified the problem definition in section 2, which involves using abductive reasoning to find a hypothesis (H) that matches an observation set $O = \{v_1, v_2, ..., v_k\}$ as closely as possible. The objective of abductive reasoning is formalized in equation (4), which uses the Jaccard score to measure the similarity between the observations and the corresponding conclusions drawn from the generated conclusions on the hidden graph.
>
> Re W2:
>
> During supervised training, the model only learns to minimize the difference between the generated and sample hypothesis sequences. However, this doesn't guarantee that the conclusion of the generated hypothesis on the testing graph will be closer to the observation, which is our ultimate objective. Additionally, this supervision signal doesn't make full use of the knowledge graph's valuable information.
>
> To address this, we implemented a PPO reward that uses the Jaccard similarity between the conclusion of the generated hypothesis on the training graph and the observation. We believe this reward provides logical information that can improve the quality of generated hypotheses by approximating our task's objective directly.
>
> The positive results in our experiments in Section 4.4 further support the value of RLF-KG in improving the quality of hypotheses generated through abductive reasoning.
>
>
> Re Q1:
>
> We confirm that both the encoder-decoder and decoder-only models we used were based on the Transformer architecture. Our main focus in this paper is to introduce the abductive reasoning task and demonstrate how the RLF-KG framework improves performance on top of a supervised trained model. Our intention is to highlight the benefits of incorporating RLF-KG into the abductive reasoning process, rather than simply comparing neural models to search-based models. While more powerful models could be used as baselines, our primary objective remains to emphasize the performance improvements achieved through the application of RLF-KG.
>
> Re Q2:
>
> In Section 4.4, we present our experimental results, which include a comparison of model performance before and after applying PPO. Table 3 shows that RLF-KG consistently improves hypothesis generation performance across three widely used datasets, using the same metric as the task's objective.
>
> Our goal is to overcome the limitations of supervised training by incorporating information from the knowledge graph. PPO is ideal for this task because it allows us to use the reward from the knowledge graph as feedback, which improves the quality of generated hypotheses by utilizing the available information. As we mentioned in our response to W2, this approach compensates for the limitations of supervised training.
>
> We chose PPO because it's a proven RL method [1] for transformer-based models.  Although there are newer methods like DPO [2] that use preference information, they're not directly applicable to our knowledge feedback tuning.
>
>
> Re Q3:
>
> The reference model is a copy of the model obtained from supervised training, and the model being fine-tuned by PPO starts off identical to the reference model. The reference model plays a crucial role in preventing excessive divergence between itself and the fine-tuned model. The KL term in the PPO objective measures this divergence and penalizes it to ensure that the models remain aligned.
>
> [1] Ouyang, Long, Jeffrey Wu, Xu Jiang, Diogo Almeida, Carroll Wainwright, Pamela Mishkin, Chong Zhang et al. "Training language models to follow instructions with human feedback." Advances in Neural Information Processing Systems 35 (2022): 27730-27744.
>
> [2] Rafailov, Rafael, Archit Sharma, Eric Mitchell, Stefano Ermon, Christopher D. Manning, and Chelsea Finn. "Direct preference optimization: Your language model is secretly a reward model." arXiv preprint arXiv:2305.18290 (2023).

---

> > ### Comment · Reviewer_dX8J · 2023-11-22
> >
> > Thanks to the author for clarifying the problems, I think the author solved my confusion very well, so I raise the rating to 6.

---

### Official Review · Reviewer_yA9N · 2023-11-08

**Soundness:** 3 good
**Presentation:** 2 fair
**Contribution:** 2 fair
**Rating:** 5
**Confidence:** 2

**Summary:**

This paper approaches the problem of abductive rule generation by proposing a supervised sequence-to-sequence model which is then fine-tuned via RL.

The goal of the paper is to, given entities, produce logical forms that describe those entities. In particular, the logical hypotheses are executed on a knowledge graph. The resulting entities are compared via Jaccard similarity to the original observations. The hypotheses themselves can be evaluated against ground-truth hypotheses using SMATCH.

The method is evaluated on 3 existing knowledge graphs. The results show that RLF-KG consistently outperforms the supervised baseline in all settings. Additionally, the proposed generation-based approach is much faster at inference than search-based methods, while having similar entity accuracy but better structural similarity to the ground truth logical forms.

**Strengths:**

The approach proposed for abductive reasoning is reasonable and was demonstrated experimentally to be accurate and fast.
The task and approach seem original, and the writing is relatively clear.
However, I am not certain about its significance.

**Weaknesses:**

The paper proposes the task of abductive logical reasoning and claims it is under-studied, but I am not convinced the task is important.
This could be improved by linking to applications, or by showing that abductive logical reasoning is a weakness in popular reasoning methods such as those that use large language models.

**Questions:**

1. I found quite a few typos and grammatical errors, but those did not hamper my understanding of the paper.
2. Can you move the results of the search baseline to the main table, table 3?

---

> ### Author Response · Authors · 2023-11-15
>
> Thank you for taking the time to review our work. We would like to respond to your comments and questions.
>
> Re W1:
>
> Abductive logical reasoning on knowledge graphs remains important because current language models, such as LLMs, struggle to reason over factual knowledge. For example, when prompted with the task of generating a logical hypothesis that matches a given observation set, ChatGPT (GPT-3.5) generated a hypothesis that claimed the devices were all first-generation Apple products, despite the observation set including an iPod touch 4th Gen. This demonstrates the model's inability to perform simple abductive reasoning.
>
> Here is the example:
> Input: “Please propose a logical hypothesis H(x) such that the set of all entities that make H(x) true should be as close to O as possible: O = {Ipad 1st Gen, Ipod touch 4th Gen, Apple TV 1st Gen}” (the 2nd example in Fig. 1)
>
> Output: “Certainly! To create a logical hypothesis H(x) that is as close to the given set O = {iPad 1st Gen, iPod touch 4th Gen, Apple TV 1st Gen} as possible, you might consider the following hypothesis:
>
> H(x): The entities in set  x  are first-generation electronic devices produced by Apple.”
>
> However, in our framework, the model is trained with feedback from the observed knowledge graph. If the knowledge graph contains basic facts about Apple's products and the model generates a faulty hypothesis, the discrepancy between the observation set and the conclusion of the hypothesis will result in a relatively low reward. This encourages the model to correct the hypothesis and improves its ability to reason over factual knowledge.
>
>
> Q1: We have fixed some typos and improved the presentation.
>
> Q2: We have moved the results of the search baseline to the main table.

---

> > ### Comment · Reviewer_yA9N · 2023-11-17
> >
> > Thanks for the response. To be clear, I think the methodology and results are solid, but I really want to be sold on the problem in the introduction.
> >
> > Concretely, this means I would like the following question answered in the introduction: Why is abductive reasoning important? Starting the story from cognitive science, causal inference, or the scientific method could help. One example that's probably too broad: if we want machines that can form generalizable and explainable theories about the world from observations, they need to be able to perform abductive reasoning [1].
> >
> > [1] Schickore, Jutta, "Scientific Discovery", The Stanford Encyclopedia of Philosophy (Winter 2022 Edition), Edward N. Zalta & Uri Nodelman (eds.).

---

> > > ### Author Response · Authors · 2023-11-18
> > >
> > > Thank you for your valuable suggestions. In response, we have incorporated further motivation for exploring abduction in the introduction, which can be summarized as follows:
> > >
> > > Abductive reasoning is a form of reasoning that is concerned with the generation of explanatory hypotheses for observed phenomena [1].
> > >
> > > This form of reasoning is a powerful tool utilized across various research domains, including
> > > -  Cognitive neuroscience: An instance of abductive reasoning is reverse inference, which is a crucial inferential strategy used to infer the most likely cognitive processes involved based on the observed brain activation patterns [2].
> > > -  Clinical diagnostics: Abductive reasoning is particularly useful in medical diagnostics, where generations of explanations have been proposed as a mechanism for studying cause-and-effect relationships in medical contexts [3].
> > >
> > > Furthermore, abductive reasoning itself is an important process in the human brain:
> > > -  It is a fundamental cognitive process that involves forming concepts, generating hypotheses, and making inferences in humans, animals, and computational machines [4].
> > >
> > > [1] https://link.springer.com/referenceworkentry/10.1007/978-1-4419-1428-6_830
> > >
> > > [2] https://link.springer.com/article/10.1007/s11229-022-03585-2
> > >
> > > [3] https://link.springer.com/referenceworkentry/10.1007/978-3-030-68436-5_13-1
> > >
> > > [4] https://link.springer.com/referencework/10.1007/978-3-030-68436-5

---

### Official Review · Reviewer_9m6p · 2023-11-09

**Soundness:** 3 good
**Presentation:** 2 fair
**Contribution:** 2 fair
**Rating:** 5
**Confidence:** 4

**Summary:**

In this paper, a data-driven approach is proposed for learning logic hypothesis based on observations and background knowledge from knowledge graphs. The key idea lies in tokenizing the hypotheses and learning the generative model from observations to hypotheses. Furthermore, reinforcement learning is utilized to allow training under the reward function about whether the observations can indeed be inferred from the learned hypotheses. Experimental results on benchmark datasets verify the effectiveness of the proposed method.

**Strengths:**

- The paper studies a very meaningful problem: learning logical hypotheses from knowledge graphs.

- The paper proposes some interesting ideas, such as tokenizing of the logical hypotheses, as well as the design of the reward function in reinforcement learning.

- The experimental results show that the proposed method indeed works.

**Weaknesses:**

- In my view, the term of abductive  reasoning is incorrectly used. The process of obtaining hypothesis based on observations and background knowledge is called induction. Abductive reasoning further requires to obtain groundings for variables in the hypothesis.

- The paper misses citation to researches on inductive logic programming (ILP), which is closely related to the problem studied in the paper. An ILP task involves learning logic programs based on logic observations and background knowledge, which is more general than reasoning on knowledge graphs. Furthermore, the term abductive learning has also be proposed before in the ILP area [1]. Related citations should be included and discussed in the paper.

- The experimental results are only quantitative. It would nicer to illustrate some qualitative examples on what kinds of hypotheses can be learned by the proposed method. For example, it would be useful to illustrate whether the proposed approach can learn lengthy hypotheses with significant complexity.

[1] Bridging Machine Learning and Logical Reasoning by Abductive Learning, NeurIPS'19.

**Questions:**

- On the bottom of Page 3, it is said that the logic clauses contains only three operations $\cup, \cap, \neg$. Does this mean that the paper only considers a limited subset of first-order logic? (e.g. propositional logic). Even though this would be enough for knowledge graphs, which can only represent relational information, clarifying the scope of learning is still necessary.

---

> ### Author Response · Authors · 2023-11-15
>
> Thank you for taking the time to review our work. We appreciate your feedback and would like to provide some clarifications regarding your comments.
>
> ## Re W1:
>
> We would like to clarify that our task differs from inductive logical programming (ILP). This is because ILP-induced logical hypotheses are general rules, as they are not grounded. In contrast, our task involves grounded logical hypotheses, which specifically describe observations rather than expressing rules. In the knowledge graph domain, inductive reasoning is equivalent to rule mining [1,2,3], discussed in Section 5.
>
> Here are some examples to illustrate the differences:
>
> Rule Mining (ILP on KG):
>
> The task involves a given knowledge graph with some positive examples regarding the relation $speak$. The goal is to find horn clause rules related to the $speak$ relation.
> $speaks(X, Y) \\leftarrow lives(X, A) \\land language(A, Y)$
> The variables in the rules are universally quantified, and the quality of the rules is measured by head coverage and precision. However, in our task, we use a real example from FB15K-237. Our observations include the following entities:
>
> $ Grant Heslov, Jason Segel, Robert Towne, Ronald Bass, Rashida Jones $
>
> Our task aims to find a logical expression based on the KG to describe these entities, rather than discovering general rules over the KG. These entities make the following logical expressions true:
>
> $V_?: Occupation(V_?, Actor) \\land Occupation(V_?, Screen Writer) \\land BornIn(V_?, Los Angeles)$
>
> This means that Grant Heslov, Jason Segel, Robert Towne, Ronald Bass, and Rashida Jones are actors and screenwriters born in Los Angeles. This logical expression is not a rule, as not all entities from the KG satisfy the logical expression.
>
> Therefore, our problem setting is distinct from inductive logical programming over knowledge graphs, specifically rule-mining.
>
> ## Re W2:
>
> We previously explained the differences between our proposed abductive KG reasoning task and ILP. Additionally, our problem setting significantly differs from the abductive learning suggested in [4]. The study in [4] uses abductive learning (ABL) to form symbolic representations via learning methods and then applies Prolog's abductive logic programming to address hand-written puzzles. These symbolic representations may not be logical expressions, and Prolog's abductive logic programming can only evaluate their truth or falsity, without generating complex first-order structured hypotheses.
>
> ## Re W3:
>
> In the appendix, we have provided several concrete examples in Tables 12, 13, and 14. These tables contain the observations, logical expressions, and their corresponding truth values. As demonstrated in the tables, our proposed method can generate logical expressions that are appropriately long, involve multiple logical connectives, and are precise (as indicated by high Jaccard scores) in describing the observations.
>
> ## Re Q1:
>
> As presented in Equations 1 and 2, we focus on a specific form of first-order logical expressions, which includes existential quantified intermediate variables and logical connectives AND/OR/NOT. We have provided further clarification regarding the scope in Section 2 of the paper.
>
>
> [1] Christian Meilicke, Melisachew Wudage Chekol, Daniel Ruffinelli, and Heiner Stuckenschmidt.
> Anytime bottom-up rule learning for knowledge graph completion. In Sarit Kraus (ed.), Proceedings of the Twenty-Eighth International Joint Conference on Artificial Intelligence, IJCAI 2019,
> Macao, China, August 10-16, 2019, pp. 3137–3143. ijcai.org, 2019. doi: 10.24963/ijcai.2019/
> 435. URL https://doi.org/10.24963/ijcai.2019/435.
>
> [2] Vinh Thinh Ho, Daria Stepanova, Mohamed H. Gad-Elrab, Evgeny Kharlamov, and Gerhard
> Weikum. Rule learning from knowledge graphs guided by embedding models. In Denny
> Vrandecic, Kalina Bontcheva, Mari Carmen Suarez-Figueroa, Valentina Presutti, Irene Celino, ´
> Marta Sabou, Lucie-Aimee Kaffee, and Elena Simperl (eds.), ´ The Semantic Web - ISWC
> 2018 - 17th International Semantic Web Conference, Monterey, CA, USA, October 8-12,
> 2018, Proceedings, Part I, volume 11136 of Lecture Notes in Computer Science, pp. 72–90.
> Springer, 2018. doi: 10.1007/978-3-030-00671-6\ 5. URL https://doi.org/10.1007/
> 978-3-030-00671-6_5.
>
> [3] Luis Galarraga, Christina Teflioudi, Katja Hose, and Fabian M. Suchanek. Fast rule mining in ´
> ontological knowledge bases with AMIE+. VLDB J., 24(6):707–730, 2015. doi: 10.1007/
> s00778-015-0394-1. URL https://doi.org/10.1007/s00778-015-0394-1.

---

> > ### Author Response · Authors · 2023-11-15
> >
> > [4] Wang-Zhou Dai, Qiu-Ling Xu, Yang Yu, and Zhi-Hua Zhou. Bridging machine learning and
> > logical reasoning by abductive learning. In Hanna M. Wallach, Hugo Larochelle, Alina
> > Beygelzimer, Florence d’Alch ́e-Buc, Emily B. Fox, and Roman Garnett (eds.), Advances in
> > Neural Information Processing Systems 32: Annual Conference on Neural Information Pro-
> > cessing Systems 2019, NeurIPS 2019, December 8-14, 2019, Vancouver, BC, Canada, pp.
> > 2811–2822, 2019. URL https://proceedings.neurips.cc/paper/2019/hash/
> > 9c19a2aa1d84e04b0bd4bc888792bd1e-Abstract.html.

---

### Comment · Area_Chair_JY6d · 2023-11-17
**Author-Reviewer Discussion Phase**

Thank you, reviewers, for your work in evaluating this submission. The reviewer-author discussion phase takes place from Nov 10-22.

If you have any remaining questions or comments regarding the rebuttal or the responses, please express them now. At the very least, please acknowledge that you have read the authors' response to your review.

Thank you, everyone, for contributing to a fruitful, constructive, and respectful review process.

AC

---

### Meta-Review · Area_Chair_JY6d · 2023-12-04

**Metareview:**

This paper proposed a data-driven approach for learning logic hypotheses based on observations and background knowledge from knowledge graphs.  Reinforcement learning is used to train the model based on a reward function that determines whether the observations can be inferred from the learned hypotheses. The effectiveness of the proposed method is confirmed through experiments conducted on benchmark datasets. This paper investigates a significant problem, namely learning logical hypotheses from knowledge graphs. The proposed method addresses the complexity of logical hypotheses by employing a generation-based approach. However, there is still a major concern that the use of the term "abduction" in this paper is inaccurate and lacks formalization.  This leads to a lack of clarity in the paper and unclear meaning of using abductive reasoning. Consequently, I recommend rejecting the paper for further improvement.

**Justification For Why Not Higher Score:**

The reason for not assigning a higher score is due to the following shortcomings identified in this paper:
1. The use of the term "abduction" in this paper is inaccurate and lacks formalization.  This leads to a lack of clarity in the paper and unclear meaning of using abductive reasoning.
2. The clarity of the paper is insufficient and the description is confusing.

**Justification For Why Not Lower Score:**

N/A

---

### Decision · Program_Chairs · 2024-01-16

Reject